# Energy Efficient Language Models through Dynamic Sparsity

## Abstract

Transformer models, despite their impressive performance, often face practical limitations due to their high computational requirements driven largely by the memory-bound KV-cache. State-space Models (SSMs) attempt to address this issue with linear attention, easing memory pressure and improving compute and memory efficiency. However, their efficiency is instead limited by dense linear layers with inherently low arithmetic intensity, again leading to a memory-bound landscape, posing challenges for deployment on hardware-constrained edge devices where these models might otherwise excel. In this work, we present a technique to induce high activation sparsity in quantized SSMs with minimal performance degradation, both for smaller-scale models suitable for edge-deployment and larger billion scale models. We nullify activations within a trainable threshold ($\pm\Delta$), which preserves outliers that are crucial for high performance. With only 1/4 of the effective MAC (Multiply-Accumulate) operations of a dense model, our sparse MatMul-free models maintain competitive performance compared to the dense base model. As GPUs offer limited support for unstructured sparsity during inference, we target a neuromorphic hardware platform that efficiently supports this dynamic and unstructured activation sparsity on a silicon level. Based on previous deployment results of a dense model, our sparsified models can increase throughput by $37\times$ while decreasing power consumption by $16\times$ compared to an edge GPU-based deployment of a comparable transformer-based LLM. Compared to a baseline dense model on the same hardware, we show improvements of $5.4\times$ in both metrics, paving the way for future explorations of highly efficient language models leveraging dynamic activation sparsity.

## 1 Introduction

The growing scale of Large Language Models (LLMs) presents significant economic and environmental challenges for inference (Fernandez et al., 2025), driven primarily by the self-attention mechanism whose cost scales quadratically with sequence length. This bottleneck makes long-context applications prohibitively expensive in resource-constrained settings. State space models (SSMs) have emerged as a powerful alternative, replacing quadratic attention with a linear-time recurrent mechanism and achieving competitive results across diverse domains (Gu & Dao, 2024; Popov et al., 2025; Smith et al., 2023; Wang et al., 2025; Voelker et al., 2019). Nevertheless, the billions of floating-point operations (FLOPs) required by SSMs still impede their deployment on edge devices, where low latency and energy efficiency are critical.

Previous model compression techniques have focused on reducing either the number of weights or activations, primarily through pruning. Weight pruning permanently removes parameters from the model, while activation pruning targets the intermediate outputs during inference. Pruning can be unstructured, removing individual weights or activations, or structured, which eliminates entire channels or blocks. While structured pruning is easier to exploit on modern specialized GPUs, the lack of fine-grain control often leads to lower model performance compared to unstructured pruning (Cheng et al., 2024). Theoretically, unstructured activation sparsity is promising because zero-valued activations can eliminate entire rows from memory access and subsequent Matrix-Vector Multiplication (MVM) operations. However, realizing benefits from unstructured sparsity is challenging for four primary reasons: 1) In long sequences, the self-attention mechanism becomes the primary memory bottleneck, diminishing any performance gains from sparsity in the dense

layers; 2) Multi-batch inference requires accessing a majority of the model's parameters unless the sparsity patterns across different input samples are closely aligned; 3) Current methods for inducing unstructured sparsity typically achieve only modest levels model-wide or else significantly degrade model performance, and 4) GPU memory interfaces (HBM) are optimized for contiguous data access and are ill-suited for the fine-grained, irregular memory patterns that result from unstructured sparsity.

SSMs inherently address the first challenge above by replacing the quadratic self-attention mechanism, thereby removing its associated memory bottleneck. In this work, we tackle the remaining issues by introducing a method to induce high activation sparsity in quantized SSMs with minimal performance degradation. We achieve this by injecting a sparsity-inducing pre-activation gate before each layer that in the forward pass, pushes activations within a learnable range of $\pm\Delta$ towards zero, while in the backward pass, it maintains a proper gradient flow for these near-zero activations, ensuring stable training. Crucially, the gate preserves high-magnitude activations (outliers), both positive and negative, which are known to be vital for LLM performance Xiao et al. (2023); Raman et al. (2025). This allows the model to maintain expressiveness without greatly disrupting the original activation distribution. This approach yields model-wide activation sparsity up to 72% with negligible impact on performance or additional training time. When targeting a neuromorphic accelerator designed to leverage such dynamic, unstructured sparsity, the benefits become substantial. We project a $24\times$ reduction in latency and a $10\times$ reduction in energy-per-token compared to a similarly sized transformer model on an edge GPU. Compared to a dense version of the same SSM on the same neuromorphic hardware, our method shows a $4.8\times$ improvement in both metrics.

## 2 RELATED WORK

Smoother non-saturating activations, such as GELU (Hendrycks & Gimpel, 2023) and SiLU/Swish (Ramachandran et al., 2017), have largely replaced ReLU due to improved optimization stability and downstream performance (Dubey et al., 2022). Unlike ReLU, these functions do not naturally produce zero activations. Recent studies in LLMs, however, show that switching back to ReLU can be done with minimal performance loss (Mirzadeh et al., 2023), with new variants like ReLU² (Zhang et al., 2024) and dReLU (Song et al., 2024) that aim to restore or enhance sparsity while retaining competitive performance.

TurboSparse (Song et al., 2024) focuses on sparsity in the feed-forward network (FFN) by introducing the dReLU activation function. The Swish activation in the SwiGLU block is replaced with ReLU, and another ReLU is added to the Up projection. By continued pre-training of these modified models, they are able to recover most of the performance of the dense baseline on benchmark tasks. The sparsity they achieve, however, is localized. Only the inputs to the Down projection are zeroed, while most other projections remain dense. As a result, even though sparsities above 90% are reported in parts of the FFN, the overall proportion of active parameters across the model is much lower.

Other approaches have tried to achieve more model-level sparsity rather than only within the FFN. Q-Sparse (Wang et al., 2024) does this by applying top-$K$ selection to activations and using a straight-through estimator to preserve gradients, combined with squared ReLU to promote sparsity. However, selecting the K largest magnitude activation requires sorting activations on a per-token basis, potentially introducing significant overheads, especially on constrained edge-hardware. Additionally, it requires synchronization across channels, complicating implementation in compute-memory integrated platforms (Pierro et al., 2025). TEAL (Liu et al., 2025) takes a different direction by introducing a layer-wise sparsification strategy. It computes token importance scores and selectively keeps only the most relevant tokens at each layer, allowing each layer to tolerate different levels of sparsity. However, this approach is only applied for the decode phase of inference, leaving the pre-fill phase fully dense and limiting potential end-to-end efficiency improvements.

## 3 BACKGROUND

### 3.1 ACTIVATION SPARSITY IN NEURAL NETWORKS

In an MVM operation, a zero-valued activation implies that all multiply-accumulate (MAC) operations involving its corresponding column of the weight matrix contribute nothing to the output. As shown in Fig. 5, this enables structured skipping: entire columns of weights associated with zero activations

can be bypassed, eliminating both the MAC operations and the need to fetch those weights from memory. Since MVMs are often memory-bound, where performance is constrained more by the cost of moving data than by raw compute throughput, reducing memory accesses can directly yield substantial energy and latency gains (Chen et al., 2019). By avoiding both the computations and memory accesses for weights corresponding to zero-valued activations, we can save bandwidth, reduce cache pressure, improve latency, and lower overall energy consumption.

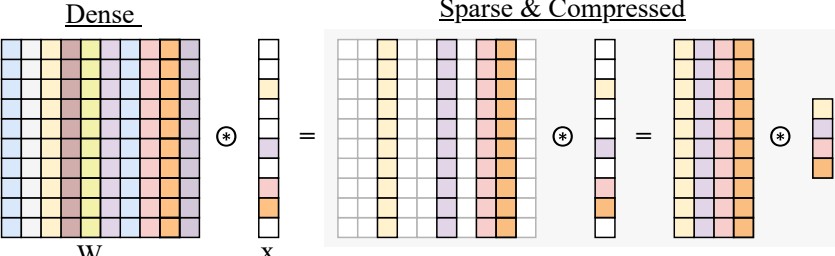

Figure 1: Illustration of activation sparsity in a matrix-vector multiplication, where zero-valued activations allow skipping associated weight accesses of entire rows in the weight matrix.

If $\rho$ denotes the fraction of zero activations, then in the ideal case, both the memory reads and the number of computations scale proportionally with $(1 - \rho)$. This means that the ideal throughput of hardware that can support sparsity is simply the dense throughput over the share of non-zero activations, i.e., $f_{\text{sparse}} = f_{\text{dense}}/(1 - \rho)$.

GPUs tend to struggle to exploit activation sparsity in inference because their architecture is built for dense, highly parallel computation with regular memory access. Unstructured sparse activations break these patterns: nonzero elements are irregularly distributed, requiring indexing and indirection that hurt coalesced memory access and reduce arithmetic efficiency. Since GPU threads execute in lockstep, skipping zeros directly is difficult without wasting compute lanes. To take advantage of sparsity, weights would need to be stored in column-major order, so that all weights linked to a nonzero activation can be fetched contiguously; however, GPUs and their libraries are optimized for dense, row-major, or block layouts, and reordering weights adds overhead. While sparse kernels can exploit activation sparsity to some extent (Liu et al., 2023), real-world speedups on GPUs are often much lower due to overheads from irregular memory access of dynamic zero-activation patterns and the limited ability of standard hardware and software to take full advantage of unstructured sparsity in a non-training setting, where the available hardware cannot be saturated with massive batch sizes.

## 3.2 LEVERAGING ACTIVATION SPARSITY IN HARDWARE ACCELERATORS

Activation sparsity has been extensively investigated in ASIC hardware due to the potential gains over GPUs, whose more regular and highly parallelized architecture cannot fully exploit it (Shi et al., 2025). Accelerators such as Eyeriss v2 (Chen et al., 2018) and SCNN (Keckler et al., 2017) primarily target convolutional networks, leveraging sparsity to reduce power consumption and increase inference throughput. More recently, neuromorphic computing has renewed interest in hardware optimized for sparse, event-driven activity (Kim et al., 2023; Sadeghi et al., 2025; Liu et al., 2022), although deployment of large models on the multi-million to billion-parameter scale required for language modeling remains limited in academic chips.

Loihi 2 (Intel Corporation, 2021) represents a state-of-the-art implementation of this approach, designed for sparse, event-based neural networks. By focusing on local event-driven computation, Loihi 2 efficiently leverages both sparse weight matrices and dynamically unstructured sparse activations using fixed-point arithmetic. In multi-chip setups, the system further leverages sparse activations through an event-driven inter chip and inter core communicating, minimizing communication overhead in by transmitting only non-zero packages.

## 4 METHODS

### 4.1 MODEL SELECTION

To meet the demands of compact models under constrained power budgets and the need for low-latency, real-time inference at the edge, various linear-attention-based quantized LLMs have been proposed (Abreu et al., 2024; 2025; Chiang et al., 2025). The MatMul-Free Language Model (MMFreeLM) (Zhu et al., 2025) pushes quantization to an extreme with ternary weights $(-1, 0, +1)$, together with low-precision 8-bit fixed-point activations, transforming dense layers into BitLinear layers that reduce multiplications to simple additions and subtractions. Based on the Gated Recurrent Unit (GRU) proposed by Cho et al. (2014), MMFreeLM uses a MMFree-Linear-GRU (MLGRU) with ternary weights, in place of the traditional self-attention mechanism in transformers. With these features combined, the MMFreeLM model has proven to be well-suited for energy-efficient inference across GPUs, edge-GPU, and neuromorphic hardware (Zhu et al., 2025; Abreu et al., 2025).

### 4.2 MOTIVATING STUDY

When extending sparsity beyond dense FFN layers to the full model, care must be taken since different linear projections vary in their role and sensitivity to pruning/sparsification (Shao et al., 2024). In SSMs, components closely tied to their linear attention, such as projections tied to the hidden state transition $h[t] \rightarrow h[t+1]$, may be especially sensitive due to their stateful nature. To study projection-wise and layer-wise sensitivity, we injected a forced sparsity into a pre-trained MMFreeLM using top-k selection of the activations with the largest magnitude, applied either per projection type (uniform across layers) or per layer (uniform across projections). Results, shown in Fig. 2, normalize loss increases by each component's share of FLOPs, highlighting where in the network inactive neurons provide the best tradeoff between active parameters and performance.

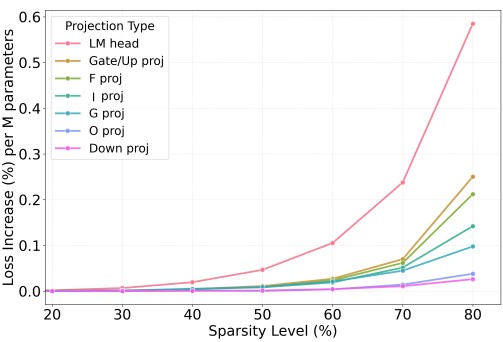
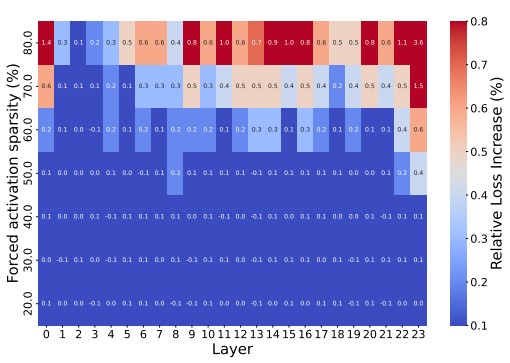

(a) Projection-wise sensitivity to forced sparsity.     (b) Layer-wise sensitivity to forced sparsity.

Figure 2: Sensitivity analysis to different degrees of forced top-k sparsity in the MMFreeLM, showing the increase in loss over the dense base-model when varying levels of sparsity are enforced. a) on a per-projection type basis and b) on a per-layer basis

**Projection-wise analysis**: The projection wise analysis shows that along with the very sensitive LM head, the projections directly involved in the next state transition $(h[t] \rightarrow h[t+1])$, $I$ and $F$, are very sensitive to enforced sparsity, whereas projections involved in the output calculation based on the current state and input $x[t]$ are less sensitive. The analysis further highlights the resilience of the large down-projection in the FFN to forced sparsity, which again aligns well with prior sparsification efforts that have targeted this projection with great success Zhang et al. (2024); Song et al. (2024). Overall, these findings align well with prior work focusing on weight pruning: in transformers, pruning attention heads leads to larger performance drops than pruning feed-forward layers (Michel et al., 2019; Voita et al., 2019), and in SSMs such as Mamba, stateful steps are much less tolerant of high sparsity than input projections or dense layers (Dao et al., 2025).

**Layer-wise analysis**: The layer-wise analysis, contrary to previous sensitivity studies on transformer-based models by Shao et al. (2024), shows that there are peaks in the sensitivity of the very first and very last layers, as well as the middle layers. This observation aligns with recent studies on

transformer architectures, which suggest that middle layers often possess greater redundancy and robustness compared to the more critical early and late layers. For example, (Ikeda et al., 2025) conducted a layer-wise importance analysis of feed-forward networks in transformer-based language models, showing that concentrating model capacity in the middle layers while reducing or removing components in the early and late layers improves downstream task performance.

## 4.3 PROPOSED SPARSIFICATION METHOD

Building on the sensitivity analysis in Section 4.2 and prior work on activation sparsity in LLMs, which showed that activations often follow Gaussian or Laplacian distributions with near-zero mean (Liu et al., 2025), we propose a sparsity-inducing pre-activation applied to the input of every linear projection. Combined with an $L_0$ surrogate loss penalty, this mechanism encourages activations to collapse toward zero, while accounting for the varying sensitivities of different projections and layers to balance task performance with activation sparsity on a model-wide level.

**Sparse per-projection pre-activation**: The proposed pre-activation is presented in equation 1 and illustrated in Fig. 3. It consists of a two-sided ReLU that zeros out activations within the range $\pm\Delta$. This preserves the overall distribution of activations by retaining both positive and negative activations, introducing only a constant offset of $\pm\Delta$ outside the zero region. The threshold $\Delta$ is treated as a learnable parameter and optimized separately for each projection during training.

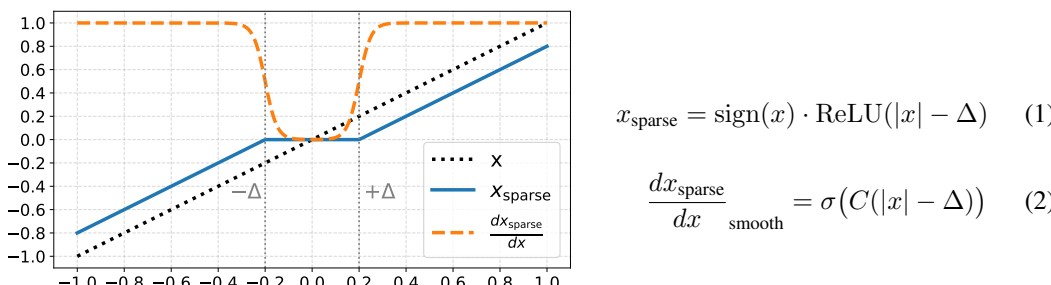

$$x_{\text{sparse}} = \text{sign}(x) \cdot \text{ReLU}(|x| - \Delta) \quad (1)$$

$$\frac{dx_{\text{sparse}}}{dx}\bigg|_{\text{smooth}} = \sigma\big(C(|x| - \Delta)\big) \quad (2)$$

Figure 3: (left) The sparse pre-activation function, with activations being zeroed out within the range $\pm\Delta$, along with the smoothed out surrogate gradient during the backwards pass. (right) The sparsity inducing pre-activation and its surrogate gradient w.r.t. x.

Similar to findings showing the smooth nature of the SiLU activation improves learning performance, especially at high levels of sparsity when a large share of gradients would be fully zeroed out by a ReLU activation (i.e. dead neurons) (Horuz et al., 2025; Dubey et al., 2022), we found that a smooth surrogate for the backwards pass, described in equation 2, gave slightly better convergence with the base models' training trajectory when compared to a hard magnitude thresholding, especially at higher sparsity levels. A slope parameter $C$ controls the steepness of the smooth mask for the derivative, with larger $C$ values pushing activations toward zero more aggressively. $C$ is fixed during training; empirically we found that $C = 20$ gives good results.

**Loss penalty and differentiable sparsity surrogate**: To encourage the model to learn to push activations to the range within $\pm\Delta$, as well as to learn an optimal value $\Delta$ on a per-projection basis, we use an $L_0$ loss penalty added to the main task loss. Since directly counting zeros in the activation vector would obstruct gradient flow to this penalty, we instead employ a surrogate sparsity measure $\hat{s}$, resulting in the following learning objective:

$$\mathcal{L} = \mathcal{L}_{\text{task}} + \lambda\,(1 - \hat{s}), \quad \hat{s} = \frac{1}{N}\sum_{i=1}^{N}\exp\big(-k\,|x_{\text{q-sparse},i}|\big), \quad (3)$$

where $\mathcal{L}_{\text{task}}$ is the primary task loss (e.g., cross-entropy), $\lambda$ controls the strength of the sparsity penalty, $N$ is the number of activations considered, $x_{\text{sparse},i}$ is the $i$-th sparse activation, $k$ is the exponential steepness parameter, and $\hat{s}$ serves as a differentiable proxy for the fraction of zero activations. Empirically, setting $k = 10$ was found to provide a good trade-off between accurately estimating true sparsity and excessively large gradient norms. The sparsity penalty is weighed on a

per-layer basis, depending on the resulting reduction in MAC operations due to a zero activation in that layer. This follows the calculations for effective MACs detailed in appendix A. Additionally, similar to previous works on neural network pruning through regularization (Wang et al., 2021), we ramp up the penalty weight term $\lambda$ slowly at the start of training, to avoid excessive sparsity before important features of the dataset have been learned. We found that a linear warm-up of 5% of the total training steps performed well.

### 4.4 DEPLOYMENT ON NEUROMORPHIC HARDWARE

#### 4.4.1 HARDWARE PLATFORM

Our hardware deployment results are derived from the real-world deployment of the dense MM-FreeLM model on the Loihi 2 platform. The platform supports two operating modes (Zhu et al., 2025), illustrated in figure 4. In pipelined mode, new inputs are introduced at every fixed time per step (TPS) and passed through successive layers, maximizing throughput. In fall-through mode, inputs are introduced only after the previous ones have been fully processed, thereby minimizing latency and allowing for a dynamically varying TPS with the per-chip workload. LLM deployment aligns naturally with these modes: prefill processing of long input sequences leverages pipelined mode for throughput efficiency, while autoregressive token generation relies on fall-through mode, since producing token $t$ must complete before token $t + 1$ can be processed.

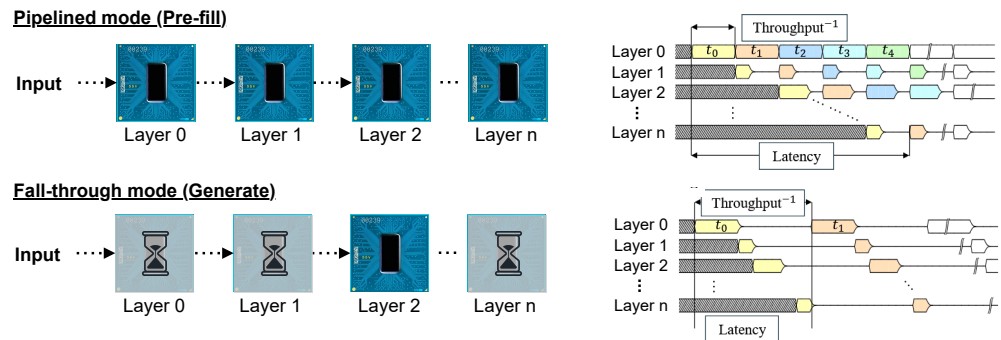

Figure 4: Different execution modes on Loihi 2, with pipelined mode (top) optimizing throughput and Fall-through mode (bottom) optimizing latency.

### 4.5 PERFORMANCE BENCHMARKING AND MODELING

We extend the performance modeling framework of Abreu et al. (2025) to account for the impact of activation sparsity. We first restate the baseline characterization of dense models and then introduce modifications that reflect the behavior of sparse activations on Loihi 2. The goal is to establish a realistic extrapolation of deployment performance under varying sparsity levels, while remaining consistent with the execution modes and architectural constraints described by Abreu et al. (2025).

**Baseline:** The MMFreeLM architecture consists of $N_{\text{blocks}}$ sequential computational blocks, each mapped to a separate Loihi 2 core or chip. For the 370M parameter model, $N_{\text{blocks}} = 24$, where each block corresponds to a layer composed of an MLGRU token-mixing unit and a ternary FFN channel-mixing block (Zhu et al., 2025).

We adopt as baseline the *measured dense multi-chip throughput* $f_{dense}^{generate}$ and $f_{dense}^{prefill}$ reported by Abreu et al. (2025). Results are provided for two inference modes: (i) *prefill*, where tokens are processed in a pipelined manner with all layers active concurrently, and (ii) *generate*, where tokens are produced autoregressively with one layer active at a time. Energy per token follows directly as $E_{\text{dense}} \propto 1/f_{\text{dense}}$, since Loihi 2 operates under an approximately constant power envelope (Abreu et al., 2025). This throughput–energy pair serves as the reference for sparse extensions.

**Impact of activation sparsity:** We define $r$ as the share of *nonzero multiply–accumulate (MAC) operations* in a block's linear layers. This reflects the effective density of executed computations,

which differs from the complement of activation sparsity because linear layers vary in size and FLOP contributions. We assume FLOPs are dominated by linear layers and ignore contributions from other operations (e.g., elementwise activations or normalization). Under these assumptions, per-block latency scales approximately linearly with $r$, consistent with prior accelerator measurements (Keckler et al., 2017; Sadeghi et al., 2025).

Activation sparsity introduces two multiplicative factors relative to this baseline:

**(i) Inter-chip communication penalty.** Dense multi-chip inference suffers a $\approx 20\%$ throughput reduction from inter-chip communication overhead (Abreu et al., 2025). Because Loihi 2 uses event-driven communication, where zero-event packets are skipped (Intel Corporation, 2021), this communication overhead can be reduced by sparse packages at the block-boundaries. We therefore model the sparsity-dependent inter-chip penalty as

$$S_{\text{comm}}(\rho_{\text{com}}) = \frac{1}{0.8 + 0.2\rho_{\text{com}}}, \tag{4}$$

where $\rho_{\text{com}}$ is the boundary activation density, and a fully dense block-boundry activation ($r = 1$) yields the throughput $f_{dense}$ reported by Abreu et al. (2025).

**(ii) MAC density speedup.** Within each block, latency scales in proportion to MAC density $r$. The intra-block factor, introduced by the zero-skipping of MAC operations, is:

$$S_{\text{MAC}}(r) = \frac{1}{r}. \tag{5}$$

**(iii) Combined throughput.** The sparse-mode throughput is obtained by multiplying the dense baseline with both factors:

$$f_{\text{sparse}} = f_{\text{dense}} \cdot S_{\text{comm}}(\rho_{\text{com}}) \cdot S_{\text{MAC}}(r). \tag{6}$$

Lastly, due to the different execution modes on Loihi 2 used during prefill and generate, the latency reduction modeled with $S_{\text{MAC}}$ varies slightly between the two modes:

- **Prefill**: Execution is pipelined across blocks, and throughput is bottlenecked by the slowest block. Let $r_{\text{max}}$ be the maximum MAC density across blocks:

$$f_{\text{sparse}}^{\text{prefill}} = f_{\text{dense}}^{\text{prefill}} \cdot \frac{1}{0.8 + 0.2\rho_{\text{com}}} \cdot \frac{1}{r_{\text{max}}}. \tag{7}$$

- **Generate**: Execution is sequential across blocks, and latency adds linearly. We approximate using the mean MAC density $r_{\text{avg}}$:

$$f_{\text{sparse}}^{\text{generate}} = f_{\text{dense}}^{\text{generate}} \cdot \frac{1}{0.8 + 0.2\rho_{\text{com}}} \cdot \frac{1}{r_{avg}}. \tag{8}$$

## 5 RESULTS

### 5.1 TRAINING SETUP

We continue training the MMFreeLM models, one sized 370M parameters for edge-deployment, and a larger 2.7B model, on 4B tokens of the FineWebEdu dataset Lozhkov et al. (2024). Training uses a cosine decaying learning rate schedule with an initial learning rate of $2e-3$ and $0.75e-3$, and a minimum learning rate of $2e-4$ and $0.75e-4$ for the 370M and 2.7B models, respectively, along with a warmup of 5% of the total 4B tokens. The initial learning rate is set to half of that used for training the original model to avoid diverging too much from the patterns learned during its original training.

For a comparison with previous methods inducing activation sparsity suitable for edge-deployment, we also train models by ReLU-fication as described by Mirzadeh et al. (2023) and the dReLU method proposed by Song et al. (2024). Additionally, we also continue-train the baseline model for the same steps as the sparse models in order to eliminate any bias introduced by the new training data. We train 3 models with our proposed sparsification method with varying degrees of enforced sparsity penalties.

## 5.2 SPARSITY OF TRAINED MODELS

We present the effective MAC of the evaluated sparsification methods compared to our proposed method, along with a per-projection type sparsity, divided into the FFN and MLGRU blocks, in table 1. Activation sparsities are captured over the entire benchmark set used in section 5.3.

Table 1: Breakdown of the activation sparsity for the MMFreeLM for different sparsity inducing methods, shown on a per-layer basis across the MLGRU and FFN.

| | MLGRU | | | | FFN | | Head | Effective MAC |
|---|---|---|---|---|---|---|---|---|
| Method | I | F | G | O | Up | Down | | sparsity ($\uparrow$)[†] |
| 370M MMFreeLM | | | | | | | | |
| Baseline (SiLU) | 1.1 | 3.2 | 1.5 | 37.3 | 1.1 | 30.1 | 2.2 | 10.0[*] (10.9) |
| ReLU | 1.1 | 2.8 | 1.4 | 37.9 | 1.2 | 87.6 | 2.2 | 23.7[*] (21.6) |
| dReLU | 1.1 | 2.8 | 1.4 | 25.0 | 1.3 | 93.2 | 2.4 | 25.1[*] (22.9) |
| **Ours** ($\lambda = 1.0$) | 45.7 | 61.2 | 48.9 | 78.2 | 63.8 | 92.2 | 24.0 | **69.5**[*] (**64.3**) |
| **Ours** ($\lambda = 2.0$) | 61.9 | 75.6 | 66.0 | 85.2 | 79.1 | 93.4 | 40.2 | **80.1**[*] (**76.2**) |
| 2.7B MMFreeLM | | | | | | | | |
| Baseline (SiLU) | 1.7 | 4.3 | 1.4 | 41.8 | 1.3 | 30.4 | 2.9 | 12.5 (12.1) |
| **Ours** ($\lambda = 1$) | 45.7 | 61.1 | 40.6 | 66.3 | 56.6 | 91.0 | 9.2 | **63.2**[*] (**61.5**) |

[*] Excluding LM Head, which is not included in the Loihi 2 implementation in Zhu et al. (2025). Value in parentheses shows the effective MAC sparsity with the LM Head included.
[†] Detailed steps for calculating the effective MAC operations, based on skippable multiplications by 0 from zero-activations, is described in Appendix A.

Due to some inherent sparsity from the fixed-point 8-bit quantization, even the base model inhibits an average baseline parameter sparsity of 10.0%. While both ReLU-based methods achieve significant levels of sparsity in the FFN at the input of the large down-projection, the impact on the overall share of active parameters is limited, as the FFN only makes up $\approx 2/3$ of total FLOPS, with the down-projection contributing to just $\approx 1/3$ of that. The result also shows that the second ReLU activation inserted with the dReLU method has a limited impact on model-wide sparsity, as the dot-product between the sparse post-activation output of the gate projection with the dense Up projection in of itself already results in a sparse vector. Our proposed method is able to achieve significantly higher levels of model-wide sparsity by not only targeting linear projections where SiLU activations can be replaced by ReLU, but all linear projections in the model.

## 5.3 PERFORMANCE ON REASONING TASKS

We evaluated the zero-shot performance of the sparsified models on the same set of language tasks as in the original MMFreeLM work, including ARC-Easy, ARC-Challenge Clark et al. (2018); Yadav et al. (2019), HellaSwag Zellers et al. (2019), OpenBookQA Mihaylov et al. (2018), PIQA Bisk et al. (2020), and WinoGrande Sakaguchi et al. (2020). The results are presented in table 2, showing a small degradation in the average reasoning task performance compared to the dense baseline model, with the 370M ($\lambda = 1$) model outperforming the dReLU model at just half the average active MAC operations.

## 5.4 ENERGY EFFICIENCY OF SPARSE MODEL

We apply the methodology described in 4.2 to the 24-chip Loihi measurements in Abreu et al. (2025) on our sparse model ($\lambda = 2$) to estimate the performance gains of a sparse model. A per-layer breakdown of the MAC density $r$ is attached in Appendix B.1. This shows a worst block MAC density $r_{max} = 0.31$ in layer 17. The input activation density of the same block is taken as the average activation sparsity to the MLGRU (i, f & g-proj) of the same layer (see (Zhu et al., 2025) for mapping details) and set to $\rho_{com} = 0.61$. Using equation 7, we calculate a decrease in latency and energy-per-token of $3.5\times$ against the dense deployment for prefill.

Table 2: Zero-shot accuracy of various sparse MMFreeLM-370M and MMFreeLM-2.7B models compared to the dense baseline model, with all models quantized to ternary weights and 8-bit activations.

| Model | Effective MACs ($\downarrow$) | ARCc | ARCe | HS | OQ | PQ | WGe | Avg. |
|---|---|---|---|---|---|---|---|---|
| | | 370M MMFreeLM | | | | | | |
| Baseline (SiLU) | 307M | 24.15 | 41.54 | 32.69 | 30.40 | 62.89 | 49.57 | 40.21 |
| ReLU | 267M | 24.95 | 39.90 | 32.99 | 31.40 | 61.70 | 49.17 | 40.02 |
| dReLU | 263M | 23.46 | 38.68 | 32.08 | 29.20 | 61.04 | 50.12 | 39.10 |
| Ours ($\lambda = 1.0$) | 118M | 23.81 | 41.29 | 31.58 | 31.00 | 61.10 | 50.59 | 39.90 |
| Ours ($\lambda = 2.0$) | 95M | 24.57 | 38.38 | 30.60 | 29.80 | 60.17 | 50.67 | 39.03 |
| | | 2.7B MMFreeLM | | | | | | |
| Baseline (SiLU) | 2.32B | 27.05 | 50.55 | 47.54 | 35.00 | 69.26 | 50.75 | 46.69 |
| Ours ($\lambda = 1$) | 1.01B | 26.71 | 48.32 | 43.43 | 35.80 | 66.43 | 52.49 | 45.53 |

Similarly, for generate, we use equation 8 with an average model-wide MAC density of $r_{avg} = 0.20$ and an average $\rho_{com} = 0.67$ to calculate an improvement in both metrics of $5.4\times$ as compared to the dense baseline. For further comparison, we also include deployment metrics by Zhu et al. (2025) of comparable Transformers models, including the 500M parameter Qwen2 model (Yang et al., 2024), and a 400M parameter Alireo model (Montebovi, 2024) on GPU and edge-GPU (Jetson).

Table 3: Throughput and efficiency across of various dense and sparse language models, including our sparse MMFreeLM, for prefill and generation across various sequence lengths, running on a NVIDIA H100 GPU, Intel's Loihi 2 and a Nvidia Jetson.

| | | | Throughput ($\uparrow$ tokens/sec) | | | | Efficiency ($\downarrow$ mJ/token) | | | |
|---|---|---|---|---|---|---|---|---|---|---|
| | Sequence length | | 500 | 1000 | 4000 | 8000 | 500 | 1000 | 4000 | 8000 |
| Generate | **MMF (sparse)** | **Loihi 2**[†] | **224.1** | **224.1** | **224.1** | **224.1** | **75.0** | **75.0** | **75.0** | **75.0** |
| | MMF (dense) | Loihi 2[*] | 41.5 | 41.5 | 41.5 | 41.5 | 405 | 405 | 405 | 405 |
| | MMF (dense) | H100[‡] | 13.4 | 13.3 | 13.5 | 13.2 | 10.1k | 10.1k | 10.0k | 9.9k |
| | TF++ | H100[‡] | 22.4 | 22.9 | 21.7 | 21.3 | 5.5k | 5.6k | 6.2k | 6.8k |
| | Alireo (400M) | Jetson[‡] | 14.3 | 14.9 | 14.7 | 15.2 | 723 | 719 | 853 | 812 |
| | Qwen2 (500M) | Jetson[‡] | 13.4 | 14.0 | 14.1 | 15.4 | 791 | 785 | 912 | 839 |
| Prefill | **MMF(sparse)** | **Loihi 2**[†] | **23.2k** | **23.2k** | **23.2k** | **23.2k** | **1.1** | **1.1** | **1.1** | **1.1** |
| | MMF (dense) | Loihi 2[*] | 6632 | 6632 | 6632 | 6632 | 3.7 | 3.7 | 3.7 | 3.7 |
| | MMF (dense) | H100[‡] | 11.4k | 13.1k | 30.6k | 51.6k | 6.1 | 5.3 | 2.5 | 1.4 |
| | TF++ | H100[‡] | 21.6k | 32.7k | 44.3k | 55.4k | 11.3 | 7.3 | 5.4 | 4.3 |
| | Alireo (400M) | Jetson[‡] | 849 | 1620 | 3153 | 2258 | 11.7 | 7.8 | 6.8 | 7.6 |
| | Qwen2 (500M) | Jetson[‡] | 627 | 909 | 2639 | 3861 | 17.9 | 13.9 | 6.7 | 4.4 |

[†] Proposed sparse model ($\lambda = 2.0$) with metrics extrapolated from dense model using on equations 7 and 8
[*] Baseline deployment results of 370M dense MMFreeLM in multi-chip setup from Abreu et al. (2025). Includes inter-chip communication slowdown over single-chip measurements.
[‡] Jeston and H100 metrics from reported deployment by Zhu et al. (2025).

# 6 CONCLUSION

In conclusion, this work introduces a novel approach to inducing high activation sparsity in an already highly compacted and efficient ternerized SSM through learnable, sparsifying pre-activations. With this method, we achieve up to 76% reduction in MAC operations, at minor performance degradation. We further demonstrate that this level of unstructured activation sparsity can yield substantial efficiency gains on hardware that supports this dynamic activation sparsity efficiently.

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

## A    DETAILED CALCULATION OF EFFECTIVE MACS ($r$)

We restrict the analysis to MVMs, as these dominate execution time compared to elementwise operations. Nonlinearities and scalar operations (e.g., inverse square root, sigmoid, bias additions) are excluded from our MAC counts, consistent with standard FLOP accounting in prior work Evci et al. (2020).

The FFN in the MMFreeLM includes a fused Up/Gate projection of size $d_h \times 2d_i$ and a Down projection of size $d_i \times d_h$. The MLGRU block contains four projections: i, f, g, & o — each of size $d_h \times d_h$. For the 370M model, the hidden size is $d_h = 1024$ and the intermediate size is $d_i = 2816$.

We define the effective MAC density $r$ as the ratio of MAC operations executed under sparsity to the number of MACs in the corresponding dense model:

$$r = \frac{\text{MAC}_{\text{sparse}}}{\text{MAC}_{\text{dense}}}.$$

Per-projection sparsity $\rho_j$ quantifies the fraction of zeros in the inputs to projection $j$. The effective MAC density for a single block is computed by summing across all projections:

$$\text{MAC}_{\text{dense}} = \sum_j d_{\text{in}}^{(j)} \cdot d_{\text{out}}^{(j)}, \quad \text{MAC}_{\text{sparse}} = \sum_j \rho_j \, d_{\text{in}}^{(j)} \cdot d_{\text{out}}^{(j)}.$$

Each nonzero activation corresponds to one row of multiplications in the projection matrix, so the number of MACs scales linearly with the activation density. Therefore, the overall effective MAC density $r$ is a *weighted average* of the per-layer densities:

$$r = \frac{\sum_j \rho_j \cdot \text{MAC}_{\text{dense}}^{(j)}}{\sum_j \text{MAC}_{\text{dense}}^{(j)}}.$$

It is worth emphasizing that because projection sizes vary across layers, $r$ is generally **not equal** to the simple arithmetic mean of per-projection sparsities:

$$r \neq \frac{1}{N} \sum_j \rho_j \equiv \rho_{\text{mean}}.$$

## B    DETAILED SPARSITY BREAKDOWN

### B.1    PER-LAYER ACTIVE-PARAMETER BREAKDOWN OF SPARSE MODEL

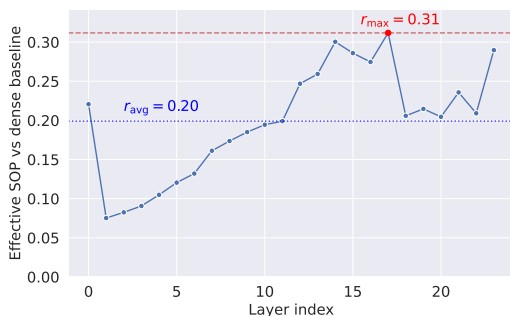

Figure 5: Effective MACs vs dense baseline of the 370M MMfreeLM model ($\lambda = 2.0$), highlighting the worst ($r_{max} = 0.31$) and average MAC density ($r_{avg} = 0.20$).

### B.2    DETAILED PROJECTION-WISE SPARSITY OF THE MODEL USED FOR THROUGHPUT/ENERGY RESULTS

| Layer | Projection | Avg. Sparsity $\rho$ | Layer | Projection | Avg. Sparsity $\rho$ | Layer | Projection | Avg. Sparsity $\rho$ |
|---|---|---|---|---|---|---|---|---|
| 0 | attn.i_proj | 0.374 | 8 | attn.i_proj | 0.615 | 16 | attn.i_proj | 0.522 |
| 0 | attn.f_proj | 0.631 | 8 | attn.f_proj | 0.751 | 16 | attn.f_proj | 0.698 |
| 0 | attn.g_proj | 0.352 | 8 | attn.g_proj | 0.686 | 16 | attn.g_proj | 0.580 |
| 0 | attn.o_proj | 0.629 | 8 | attn.o_proj | 0.879 | 16 | attn.o_proj | 0.873 |
| 0 | mlp.gate_proj | 0.888 | 8 | mlp.gate_proj | 0.830 | 16 | mlp.gate_proj | 0.687 |
| 0 | mlp.down_proj | 0.972 | 8 | mlp.down_proj | 0.955 | 16 | mlp.down_proj | 0.885 |
| 1 | attn.i_proj | 0.864 | 9 | attn.i_proj | 0.570 | 17 | attn.i_proj | 0.526 |
| 1 | attn.f_proj | 0.926 | 9 | attn.f_proj | 0.728 | 17 | attn.f_proj | 0.713 |
| 1 | attn.g_proj | 0.888 | 9 | attn.g_proj | 0.678 | 17 | attn.g_proj | 0.573 |
| 1 | attn.o_proj | 0.935 | 9 | attn.o_proj | 0.883 | 17 | attn.o_proj | 0.885 |
| 1 | mlp.gate_proj | 0.907 | 9 | mlp.gate_proj | 0.824 | 17 | mlp.gate_proj | 0.608 |
| 1 | mlp.down_proj | 0.991 | 9 | mlp.down_proj | 0.943 | 17 | mlp.down_proj | 0.867 |
| 2 | attn.i_proj | 0.840 | 10 | attn.i_proj | 0.524 | 18 | attn.i_proj | 0.624 |
| 2 | attn.f_proj | 0.891 | 10 | attn.f_proj | 0.703 | 18 | attn.f_proj | 0.741 |
| 2 | attn.g_proj | 0.859 | 10 | attn.g_proj | 0.669 | 18 | attn.g_proj | 0.659 |
| 2 | attn.o_proj | 0.955 | 10 | attn.o_proj | 0.861 | 18 | attn.o_proj | 0.912 |
| 2 | mlp.gate_proj | 0.902 | 10 | mlp.gate_proj | 0.827 | 18 | mlp.gate_proj | 0.785 |
| 2 | mlp.down_proj | 0.993 | 10 | mlp.down_proj | 0.931 | 18 | mlp.down_proj | 0.898 |
| 3 | attn.i_proj | 0.806 | 11 | attn.i_proj | 0.513 | 19 | attn.i_proj | 0.623 |
| 3 | attn.f_proj | 0.864 | 11 | attn.f_proj | 0.697 | 19 | attn.f_proj | 0.743 |
| 3 | attn.g_proj | 0.836 | 11 | attn.g_proj | 0.633 | 19 | attn.g_proj | 0.628 |
| 3 | attn.o_proj | 0.946 | 11 | attn.o_proj | 0.860 | 19 | attn.o_proj | 0.912 |
| 3 | mlp.gate_proj | 0.901 | 11 | mlp.gate_proj | 0.827 | 19 | mlp.gate_proj | 0.770 |
| 3 | mlp.down_proj | 0.992 | 11 | mlp.down_proj | 0.931 | 19 | mlp.down_proj | 0.901 |
| 4 | attn.i_proj | 0.773 | 12 | attn.i_proj | 0.467 | 20 | attn.i_proj | 0.635 |
| 4 | attn.f_proj | 0.832 | 12 | attn.f_proj | 0.674 | 20 | attn.f_proj | 0.753 |
| 4 | attn.g_proj | 0.808 | 12 | attn.g_proj | 0.596 | 20 | attn.g_proj | 0.632 |
| 4 | attn.o_proj | 0.928 | 12 | attn.o_proj | 0.610 | 20 | attn.o_proj | 0.919 |
| 4 | mlp.gate_proj | 0.892 | 12 | mlp.gate_proj | 0.794 | 20 | mlp.gate_proj | 0.782 |
| 4 | mlp.down_proj | 0.988 | 12 | mlp.down_proj | 0.912 | 20 | mlp.down_proj | 0.910 |
| 5 | attn.i_proj | 0.730 | 13 | attn.i_proj | 0.438 | 21 | attn.i_proj | 0.672 |
| 5 | attn.f_proj | 0.809 | 13 | attn.f_proj | 0.662 | 21 | attn.f_proj | 0.767 |
| 5 | attn.g_proj | 0.784 | 13 | attn.g_proj | 0.578 | 21 | attn.g_proj | 0.615 |
| 5 | attn.o_proj | 0.915 | 13 | attn.o_proj | 0.649 | 21 | attn.o_proj | 0.916 |
| 5 | mlp.gate_proj | 0.879 | 13 | mlp.gate_proj | 0.775 | 21 | mlp.gate_proj | 0.711 |
| 5 | mlp.down_proj | 0.983 | 13 | mlp.down_proj | 0.903 | 21 | mlp.down_proj | 0.902 |
| 6 | attn.i_proj | 0.704 | 14 | attn.i_proj | 0.443 | 22 | attn.i_proj | 0.730 |
| 6 | attn.f_proj | 0.806 | 14 | attn.f_proj | 0.662 | 22 | attn.f_proj | 0.812 |
| 6 | attn.g_proj | 0.762 | 14 | attn.g_proj | 0.522 | 22 | attn.g_proj | 0.667 |
| 6 | attn.o_proj | 0.916 | 14 | attn.o_proj | 0.527 | 22 | attn.o_proj | 0.923 |
| 6 | mlp.gate_proj | 0.865 | 14 | mlp.gate_proj | 0.721 | 22 | mlp.gate_proj | 0.736 |
| 6 | mlp.down_proj | 0.976 | 14 | mlp.down_proj | 0.891 | 22 | mlp.down_proj | 0.912 |
| 7 | attn.i_proj | 0.633 | 15 | attn.i_proj | 0.453 | 23 | attn.i_proj | 0.790 |
| 7 | attn.f_proj | 0.761 | 15 | attn.f_proj | 0.667 | 23 | attn.f_proj | 0.860 |
| 7 | attn.g_proj | 0.701 | 15 | attn.g_proj | 0.524 | 23 | attn.g_proj | 0.625 |
| 7 | attn.o_proj | 0.894 | 15 | attn.o_proj | 0.844 | 23 | attn.o_proj | 0.883 |
| 7 | mlp.gate_proj | 0.842 | 15 | mlp.gate_proj | 0.694 | 23 | mlp.gate_proj | 0.549 |
| 7 | mlp.down_proj | 0.965 | 15 | mlp.down_proj | 0.886 | 23 | mlp.down_proj | 0.917 |
| - | - | - | - | - | - | - | lm_head | 0.402 |

Table 4: Per-layer average sparsity of the sparse $\lambda = 1.5$ model.

