# OpenReview forum: "Energy Efficient Language Models through Dynamic Sparsity"
_ICLR.cc/2026/Conference — Submitted to ICLR 2026_

### Official Review · Reviewer_JPiG · 2025-10-26

**Soundness:** 3
**Presentation:** 3
**Contribution:** 2
**Rating:** 4
**Confidence:** 2

**Summary:**

This paper proposes a novel method to induce high activation sparsity in quantized SSMs (MMFreeLM).

**Strengths:**

1. The paper's experiments show that this method can achieve 76% reduction in effective MAC  operations with negligible impact on task performance.
2. The method is easy but useful. Using a learnable, per-projection threshold $\Delta$ allows the model to assign different sparsity levels based on layer sensitivity.

**Weaknesses:**

1. The paper's most impactful claims (e.g., 75mJ/token and 224.1 throughput in Table 3) do not appear to be actual measured results from running the sparse model on Loihi 2. As described in Section 4.5, these figures are calculateed from a performance model based on measurements of the dense model. The paper's claims would be strengthened if the authors could provide real acceleration data from Loihi 2, even on a single, small-scale sparse layer.

2. The proposed method requires keep training on top of the pre-trained model. This process introduces additional learnable parameters and a new loss term. The paper does not discuss the additional computational overhead or convergence time this adds to the training phase. Could author discuss and add those cost?

**Questions:**

See weakness above.

---

> ### Author Response · Authors · 2025-11-21
> **Response to reviewer JPiG**
>
> Thank you for your comments. We have attempted to address your concerns and answer your questions below.
>
> ___
>
> **Q1: The paper's most impactful claims (e.g., 75mJ/token and 224.1 throughput in Table 3) do not appear to be actual measured results from running the sparse model on Loihi 2.**
>
> **A1:** (Copied from response to reviewers with similar concerns) While this is a valid concern, we believe our performance modeling of the gains from deploying a sparse model versus a dense model is well supported by similar existing published academic architectures [1][2], which provide sound performance analyses of sparsity versus throughput, as well as real-world baseline measurements [3] and Intel's own architectural paper [4] describing the similar microarchitecture of Loihi1.
>
> We would like to further strengthen this claim by adding reference [5], where Appendix Table 3 shows the scaling of reconstructions per second (i.e., throughput) on Loihi2 against fixed thresholds 𝜆, which relate to sparsity in Figure 4. From 𝜆 = 2^-6 (20% sparsity) to 𝜆 = 2^-1 (83% sparsity), throughput increases by 4.5x. This closely aligns with our model’s assumption that Loihi2 performance scales linearly with the reduction in MAC operations, which predicts a performance increase of 4.7x (i.e.,(1–0.2)/(1–0.83)). Although the model architecture used in [5] differs from ours, the key observation that throughput increases nearly proportionally with sparsity (or effective reduction in MACs) remains valid in our opinion.
> ___
>
> **Q2: The proposed method requires keep training on top of the pre-trained model. This process introduces additional learnable parameters and a new loss term. The paper does not discuss the additional computational overhead or convergence time this adds to the training phase. Could author discuss and add those cost?**
>
> **A2:** While the method does require continue-training, the models quickly converge towards the original model's performance on benchmark tasks. For the 2.7B model, we trained on just 2B tokens, which is an overhead of just 2% additional training tokens compared to the original models 100B training tokens. For the smaller 370M model, the overhead was around 13% additional training tokens, but since this model is small enough to be trained even on a single GPU, we did not deem this very significant.
>
> Additionally, the added sparsity surrogate (eq. 3) is applied as a element-wise operation, resulting in relatively few OPs compared to the actual linear layer. While this operation does include a exponential, these are typically highly optimized and accelerated on modern GPUs through [SFUs](https://modal.com/gpu-glossary/device-hardware/special-function-unit).
>
> ___
>
> [1] Stephen W. Keckler, David Burger, Hadi Esmaeilzadeh, et al. Scnn: An accelerator for compressed- sparse convolutional neural networks. Proceedings of the 44th Annual International Symposium on Computer Architecture
>
> [2] M. Sadeghi et al., "NEXUS: A 28nm 3.3pJ/SOP 16-Core Spiking Neural Network With a Diamond Topology for Real-Time Data Processing," in IEEE Transactions on Biomedical Circuits and Systems
>
> [3] Steven Abreu et al., Neuromorphic principles for efficient large language models on intel loihi 2
>
> [4] G. Orchard et al., "Efficient Neuromorphic Signal Processing with Loihi 2," 2021 IEEE Workshop on Signal Processing Systems (SiPS)
>
> [5] Gavin Parparat et al., Implementing and Benchmarking the Locally Competitive Algorithm on the Loihi 2 Neuromorphic Processor, Proceedings of the 2023 International Conference on Neuromorphic SysteICONS23

---

### Official Review · Reviewer_R6HG · 2025-10-27

**Soundness:** 3
**Presentation:** 3
**Contribution:** 2
**Rating:** 6
**Confidence:** 3

**Summary:**

This paper proposes a method to induce high activation sparsity in quantized State Space Models. The approach uses learnable threshold-based pre-activation gates that zero out activations within a delta while preserving outliers. The authors demonstrate that their sparse models maintain competitive performance with up to 72% activation sparsity and project significant efficiency gains when deployed on neuromorphic hardware.

**Strengths:**

1. The sensitivity analysis provides valuable insights into which projections and layers can tolerate sparsity, and provides proper motivation for the paper. The authors offer an orthogonal complement to pruning and quantisation, while building upon existing techniques.
2. The paper evaluates both 370M and 2.7B parameter models across multiple benchmarks, showing consistent results.
3. The learnable two-sided ReLU with smooth surrogate gradient is computationally lightweight and unlike prior ReLU-based sparsification it generalises across all projections and learns per-projection thresholds.
3. Targeting neuromorphic hardware that can actually exploit unstructured sparsity is a good choice that was thoroughly explained.

**Weaknesses:**

1.  Results are extrapolated from dense deployments rather than measured on actual sparse models. Real deployment results on Loihi 2 would strengthen credibility.
2. Competing SSM-based efficiency techniques (Mamba pruning, LoRA-style compression, structured token pruning) are not directly compared experimentally.
3. The paper claims “minimal additional training cost” but does not report training time or compute the overhead introduced by sparsity regularisation.
3.  The paper can include more important prior work on activation sparsity for neuromorphic computing. demonstrates similar sparsity-inducing techniques for neuromorphic hardware on simpler models. [ Activity Sparsity Complements Weight Sparsity for Efficient RNN Inference (https://arxiv.org/abs/2311.07625), Sparsity-Aware Hardware-Software Co-Design of Spiking Neural Networks: An Overview (https://arxiv.org/abs/2408.14437)
]

**Questions:**

1. Were the sparsity thresholds initialised globally or per-layer? How sensitive are results to this initialisation?
2. Have you explored structured sparsity variants such as block or channel for compatibility with GPU inference? Were the more stringent sparsity setups too detrimental to results achieved?
3. How stable is training with high lambda, beyond what values are there cases where too much sparsity leads to collapse?

---

> ### Author Response · Authors · 2025-11-24
> **Response to Reviewer R6HG (1/2)**
>
> Thank you for the thorough review and input. We have attempted to address your questions and concerns in the text below.
>
> ___
>
> **Q1: Were the sparsity thresholds initialised globally or per-layer? How sensitive are results to this initialisation?**
>
> **A1:** We initialized the per-layer thresholds to zero across all layers. While we initially tried initializing the thresholds based on the sensitivity analysis carried out in section 4.2, we found that the model converged on the same thresholds without the initialization, and that carrying out this pre-training step each time just unnecessarily complicated the training process.
>
> ___
>
> **Q2: Have you explored structured sparsity variants such as block or channel for compatibility with GPU inference?**
>
>
> **A2:** Our focus has mostly been on showing the advantages of fully unstructured/dynamic activation sparsity on less conventional hardware that offers support unmet by GPUs.  We do however believe that the method, due to its very high overall activation sparsity, is well suited for being extended to block-wise sparsity (e.g 2:4 sparsity), where the constraint could be applied to our sparse models and the model can be further tuned with this enforced constraint. However, in MVM operations, zeroes in the activation-vector already lead to somewhat structured sparsity in the memory access, as an entire column of the weight matrix can skip to be read from memory. We are currently investigating the feasibility of a custom triton kernel that better supports this.
>
> ___
>
> **Q3: Were the more stringent sparsity setups too detrimental to results achieved? (+How stable is training with high lambda, beyond what values are there cases where too much sparsity leads to collapse?)**
>
>
> **A3:** We chose not to include models with a higher penalty than λ=2, as we deemed that this gave an attractive accuracy/throughput trade-off compared to previous SOTA methods. We have uploaded supplementary plots showing the training trajectory for different degrees of sparsification (λ=1,2,3) for the 370M mmfreelm model, showing a significant increase in the evaluation loss as λ increases beyond 2. Additionally, the gradient norm for the more aggressively sparsified models increases quite significantly, potentially leading to unstable training.
> Lastly, we made the assumption that at excessive sparsity levels, other constant operations that do not scale with sparsity, such as normalization and activations, would begin to dominate the runtime even on optimized hardware such as Loihi2.
>
> ___

---

> > ### Author Response · Authors · 2025-11-24
> > **Response to Reviewer R6HG (2/2)**
> >
> > We have additionally attempted to address your raised weaknesses about this work below.
> > ___
> >
> > **Results are extrapolated from dense deployments rather than measured on actual sparse models. Real deployment results on Loihi2 would strengthen credibility.**
> >
> >  While we would have liked to include real world deployment of the sparse model, we feel that our methodology for extending the data from the base models real-world deployment with a highly accurate performance model, that extends this real world data to include activation sparsity, is reasonable based on known information about the hardware architecture. This is partly supported by similar existing published academic architectures [1][2], which provide sound performance analyses of sparsity versus throughput, as well as real-world baseline measurements [3] and Intel's own architectural paper [4] describing the similar microarchitecture of Loihi1.
> >
> > We would like to further strengthen this claim by adding reference [5], where Appendix Table 3 shows the scaling of reconstructions per second (i.e., throughput) on Loihi2 against fixed thresholds 𝜆, which relate to sparsity in Figure 4. From 𝜆 = 2^-6 (20% sparsity) to 𝜆 = 2^-1 (83% sparsity), throughput increases by 4.5x. This closely aligns with our model’s assumption that Loihi2 performance scales linearly with the reduction in MAC operations, which predicts a performance increase of 4.7x (i.e.,(1–0.2)/(1–0.83)). Although the model architecture used in [5] differs from ours, the key observation that throughput increases nearly proportionally with sparsity (or effective reduction in MACs) remains valid in our opinion.
> >
> > [1] Stephen W. Keckler, David Burger, Hadi Esmaeilzadeh, et al. Scnn: An accelerator for compressed- sparse convolutional neural networks. Proceedings of the 44th Annual International Symposium on Computer Architecture
> >
> > [2] M. Sadeghi et al., "NEXUS: A 28nm 3.3pJ/SOP 16-Core Spiking Neural Network With a Diamond Topology for Real-Time Data Processing," in IEEE Transactions on Biomedical Circuits and Systems
> >
> > [3] Steven Abreu et al., Neuromorphic principles for efficient large language models on intel loihi 2
> >
> > [4] G. Orchard et al., "Efficient Neuromorphic Signal Processing with Loihi 2," 2021 IEEE Workshop on Signal Processing Systems (SiPS)
> >
> > [5] Gavin Parparat et al., Implementing and Benchmarking the Locally Competitive Algorithm on the Loihi 2 Neuromorphic Processor, Proceedings of the 2023 International Conference on Neuromorphic SysteICONS23
> >
> > ___
> >
> > **The paper claims “minimal additional training cost” but does not report training time or compute the overhead introduced by sparsity regularisation.**
> >
> > While the method does require continue-training, the models quickly converge towards the original model's performance on benchmark tasks. For the 2.7B model, we trained on just 2B tokens, which is an overhead of just 2% additional training tokens compared to the original models 100B training tokens. For the smaller 370M model, the overhead was around 13% additional training tokens, but since this model is small enough to be trained even on a single GPU, we did not deem this very significant. Additionally, the added sparsity surrogate (eq. 3) is applied as a element-wise operation, resulting in relatively few OPs compared to the actual linear layer. While this operation does include a exponential, these are typically highly optimized and accelerated on modern GPUs through SFUs.

---

> > > ### Comment · Reviewer_R6HG · 2025-11-24
> > > **Response**
> > >
> > > Thank you for clarifying my doubts, however I feel I have given a high enough score and wont be changing it.

---

### Official Review · Reviewer_TZy8 · 2025-10-28

**Soundness:** 3
**Presentation:** 3
**Contribution:** 2
**Rating:** 2
**Confidence:** 3

**Summary:**

This paper introduces a sparsification algorithm for SSMs targeting neuromorphic hardware, namely Loihi 2.

**Strengths:**

The sparsification algorithm of the paper seems to perform well, and introduces significant activation sparsity.

**Weaknesses:**

Sparsification is an old topic and I find the paper's contribution to be very limited to a single proposal that was laid out in one figure. They then introduced a modification to the loss function to "encourage" the pushing of the values to +/- \Delta. There was no explanation why the function is a good one, and also no details about how \Delta is learned - how is \Delta updated in the training process?

The other major weakness of the paper is that it just evaluated its proposal with other activation functions. Since sparsification is not new, there ought to be an evaluation against other sparsification methods. Without this, it is hard to place the contribution of the work.

**Questions:**

1. How would your method compare to other state-of-the-art sparsification algorithms, even for traditional ANNs?

2. Any reason for k = 10 (page 5) being a "good" value? Also, what's the intuition behind Eq. 3?

3. Would the approach also work for transformers?

**Details Of Ethics Concerns:**

None.

---

> ### Author Response · Authors · 2025-11-25
> **Response to reviewer TZy8**
>
> Thank you for review and valuable feedback. We have attempted to address your concerns and answer your questions below.
> ___
>
> **Q1: How would your method compare to other state-of-the-art sparsification algorithms, even for traditional ANNs?**
>
> **A1:** We have detailed a comparison to, in our opinion, the most comparable works that introduce **dynamic** activation sparsity without requiring sorting & syhcnroinization (as in top-k approaches) or fixed pruning (i.e. permanently disabling entire neurons). As we demonstarted, both ReLU-ficiation and the turbosparse approach are only able to reduce model-wide MAC by around 25% in our evaluation (see table 1), as the ReLU activations applied only introduce zero-activations at the input of the down-projection. This greatly limits the real-world gains when deployed on sparsity-supporting hardware. Top-k based approaches, such as Q-sparse, only achieve 46–58% MAC sparsity at accuracy degradation comparable to ours. More importantly however for our deployment target, because their approach relies on top-k masking, it requires sorting of activations based on magnitude, which adds a performance overhead and limits efficiency on platforms such as Loihi2, as it would require synchronization between cores when the workload of a single layer is distributed amongst multiple cores. In contrast, our method uses a simple pre-activation scheme implemented with simple set of comparisons.
>
> Our work aims to demonstrate the benefits of dynamic activation sparsity, where a subset of activations is selectively zeroed during the forward pass without pruning weights, using a learnable threshold mechanism. We focused comparisons on standard activation functions to isolate and clearly show the impact of the proposed mechanism itself. We acknowledge that sparsification is a broad field, and techniques such as structured pruning, MoEs, or token-level pruning operate at different levels and are largely complementary. Evaluating against these methods could provide additional insights, but it was beyond the scope of the current work. We view our method as compatible with such approaches, and integrating and comparing with other dynamic, activation-level sparsity techniques is an important direction for future work.
>
> ___
>
> **Q2:Any reason for k = 10 (page 5) being a "good" value? Also, what's the intuition behind Eq. 3?**
>
> **A2:** Eq. 3 provides a differentiable proxy for activation sparsity. Directly penalizing the number of nonzero activations is non-differentiable, so we instead use a smooth approximation that approaches 0 for large activation values and 1 for small ones. The regularization term encourages more activations to be close to zero while preserving differentiability, allowing gradient-based optimization to naturally reduce small activations and push them towards the "zeroing-out" range of +-Δ.
>
> The parameter k controls the steepness of the exponential. Higher k makes the surrogate closer to the true sparsity function (more “exact”), but can reduce training stability. Lower k smooths the penalty, improving stability at the cost of a looser approximation. We chose \(k = 10\) empirically as a balance between sparsity accuracy and stable training but found that values close to 10 also gave similar results. We found that much higher values lead to large gradient norms (> 10) and generally unstable training.
> ___
>
> **Q3: Would the approach also work for transformers?**
>
> **A3:** We have no reason to believe this approach would be unsuitable for transformers, particularly in their dense feed-forward networks, where methods such as Turbo-Sparse have already been applied. However, during transformer inference, the quadratic scaling of the compute cost of self-attention with context length causes these operations to dominate, resulting in the MAC operations from the linear projections to become comparatively insignificant . As a result, skipping MACs through activation sparsity has a limited effect on the overall compute cost. For this reason, we focus on SSMs, where regardless of context length, the linear layers tend to dominate the compute workload and a significant end-to-end performance gains can be achieved through dynamic activation sparsity, as demonstrated by this work.
>
> ___
>
> Furthermore, you raised the need to clarify **how is Δ is learned and how is it updated in the training process?**
>
> **A:** The sparsificaiton threshold Δ is trained jointly with the model weights using standard backpropagation, with gradients computed through a differentiable surrogate of the hard threshold. The training loss combines the task loss with a sparsity-promoting term that penalizes nonzero activations, providing a gradient signal that encourages Δ to increase where more sparsity is desired and decrease where preserving activations is important for performance. This allows each layer to adaptively learn the appropriate activation threshold, balancing sparsity and task accuracy.

---

### Official Review · Reviewer_GaZW · 2025-11-04

**Soundness:** 2
**Presentation:** 2
**Contribution:** 2
**Rating:** 4
**Confidence:** 3

**Summary:**

To address the high computational requirements introduced by the memory-bound KV-cache, State-space Models (SSMs) are proposed to ease memory pressure. But the new bottleneck of SSMs (linear layers) is still memory-bound, making it hard to be deployed on resource-constrained edge devices.
This paper proposed a trainable dynamic-sparsity mechanism for quantized State-Space Models (SSMs). The sparse model can achieve 37x better throughput on Intel Loihi 2 when compared with the dense model on edge GPU & 5.4x better throughput when compared with the dense model on the same hardware.

**Strengths:**

Over 60% MAC sparsity can be achieved with ~1% accuracy loss.

**Weaknesses:**

Loihi 2 sparse results are modeled, not measured.
Novelty compared with other work? It seems both TurboSparse & Q-Sparse have already introduced activation sparsity.

**Questions:**

How is the paper compared with other sparse LLM work?
What is the extra parameter size for the sparsification (delta, etc)?

---

> ### Author Response · Authors · 2025-11-21
> **Response to Reviewer GaZW (1/2)**
>
> Thank you for review and valuable feedback. We have attempted to address your concerns and answer your questions below.
>
> ___
>
> **Q1: Novelty compared with other work?**
>
> **A1:** We believe the novelty in our method lies mainly in the way we are able to achieve **model-wide sparsity** across all the models' linear layers, not just in the parts of the FFN as turbosparse or simple ReLu-ficiation is able to. This gives a significantly greater reduction of total MAC operations, with upwards of 80% of total MACs being skipped. The turbosparse approach is only able to reduce model-wide MAC by around 25% in our evaluation (see table 1), as the ReLU activations they apply only introduce zero-activations at the input of the down-projection.  This greatly limits the real-world gains when deployed on sparsity-supporting hardware.
>
> While Q-sparse targets activation sparsity globally using top-k masks, it achieves only 46–58% MAC sparsity at levels of accuracy degredation comparable to our method. More importantly however for our deployment target, because their approach relies on top-k masking, it requires sorting all activations in each layer, which introduces a performance overhead and limits hardware efficiency on platforms such as Loihi2, as it would require synchronization between cores when the workload of a single layer is distributed amongst multiple cores. In contrast, our method uses a simple pre-activation scheme implemented with simple set of comparisons.
>
> ___
>
> **Q2: How is the paper compared with other sparse LLM work?**
>
> **A2:** Similar to the comparison carried out in the Q-sparse work, we have compared to ReLUfication along with the method proposed in turbosparse, as this most closely aligned with our aim of inducing **dynamic** activation sparsity purely from simple pre/post-activation functions. For this reason, we did not see a comparison to Q-sparse as strictly relevant with our work and deployment target, as their method of enforcing activation sparsity is not suitable for the hardware target we chose.
>
> We focused our comparisons on standard activation functions in order to clearly demonstrate the impact of the proposed mechanism itself. We agree that sparsification is a broad field, and techniques such as structured pruning, MoEs, or token-level pruning operate at different levels and are largely complementary. Evaluating against these methods could provide additional insights, but it was beyond the scope of the current work. We view our method as compatible with such approaches, and we see integrating and comparing with other dynamic, activation-level sparsity techniques as an important direction for future work.
>
> ___
> **Q3: What is the extra parameter size for the sparsification (Δ, etc)?**
>
> **A3:** The extra introduced Δ parameter is learned on a per-projection basis. Across the model, this adds just a handful of scalar parameters per block, which we deemed negligible compared to the weights of the linear layer. The overall overhead on the training is negligible in terms of parameter count, and the added sparsity surrogate (eq. 3) is applied as a element-wise operation, resulting in relatively few OPs compared to the actual linear layer.

---

> ### Author Response · Authors · 2025-11-25
> **Response to Reviewer GaZW (2/2)**
>
> ___
> **Q4: Loihi 2 sparse results are modeled, not measured**
>
> **A4:** While this is a valid concern, we believe our performance modeling of the gains from deploying a sparse model versus a dense model is well supported by similar existing published academic architectures [1][2], which provide sound performance analyses of sparsity versus throughput, as well as real-world baseline measurements [3] and Intel's own architectural paper [4] describing the similar microarchitecture of Loihi1.
>
> **We would like to further strengthen this claim by adding reference [5]**, where Appendix Table 3 shows the scaling of reconstructions per second (i.e., throughput) on Loihi2 against fixed thresholds 𝜆, which relate to sparsity in Figure 4. From 𝜆 = 2^-6 (20% sparsity) to 𝜆 = 2^-1 (83% sparsity), throughput increases by 4.5x. This closely aligns with our model’s assumption that Loihi2 performance scales linearly with the reduction in MAC operations, which predicts a performance increase of 4.7x (i.e.,(1–0.2)/(1–0.83)). Although the model architecture used in [5] differs from ours, the key observation that throughput increases nearly proportionally with sparsity (or effective reduction in MACs) remains valid in our opinion.
>
> ___
>
> [1] Stephen W. Keckler, David Burger, Hadi Esmaeilzadeh, et al. Scnn: An accelerator for compressed- sparse convolutional neural networks. Proceedings of the 44th Annual International Symposium on Computer Architecture
>
> [2] M. Sadeghi et al., "NEXUS: A 28nm 3.3pJ/SOP 16-Core Spiking Neural Network With a Diamond Topology for Real-Time Data Processing," in IEEE Transactions on Biomedical Circuits and Systems
>
> [3] Steven Abreu et al., Neuromorphic principles for efficient large language models on intel loihi 2
>
> [4] G. Orchard et al., "Efficient Neuromorphic Signal Processing with Loihi 2," 2021 IEEE Workshop on Signal Processing Systems (SiPS)
>
> [5] Gavin Parparat et al., Implementing and Benchmarking the Locally Competitive Algorithm on the Loihi 2 Neuromorphic Processor, Proceedings of the 2023 International Conference on Neuromorphic SysteICONS23

---

### Meta-Review · Area_Chair_E2Ds · 2025-12-17

**Summary:**

This paper introduces a method for inducing high activation sparsity in quantized State-space Models (SSMs), specifically targeting deployment on neuromorphic hardware (Intel Loihi 2). The main method is a learnable threshold mechanism to zero out activations with in a range. This method achieves high sparsity with a minimal quality loss.

Common concerns and my takes after reading the rebuttal:

- The performance gains on Loihi 2 are derived from a performance model rather than physical measurements. The authors provided references to support their modeling assumptions in the rebuttal. However, 1) recent works on activation sparsity such as TurboSparse often report practical speedup numbers which were critical to validate the efficiency benefits, 2) if such measurements are not to be provided, then I'd expect the paper to excel more on other aspects, e.g., evaluation of their activation sparsity technique more thoroughly in comparison with related work, which leads to the next common concern.

- A few reviewers questioned a lack of comparative baselines, noting that activation sparsity is a well-established field. The authors argued that their study is already detailed in comparing with the "most comparable works", which ruled out many of the popular methods. However, a) it can still be interesting to include those methods to understand where the proposed method stands, b) some of the claims are not entirely true. Specifically, 1) while ReLU-ficiation and turbosparse only reduces model-wide MAC by around 25%, there are methods [a] where sparsity was introduced to other parts of the model as well for more sparsity; 2) for issues concerning sorting in top-k, there are approximate versions [b] where sorting is not required.

[a] [Sparse is Enough in Scaling Transformers](https://arxiv.org/pdf/2111.12763)

[b] [Spark Transformer: Reactivating Sparsity in FFN and Attention](https://arxiv.org/pdf/2506.06644)

Overall, while the direction of combining SSMs with neuromorphic hardware is promising, the current submission relies on theoretical performance modeling rather than empirical hardware validation. Furthermore, the novelty of the sparsification mechanism itself was not sufficiently distinguished from prior art, and the evaluation is quite limited compared to existing studies. The authors may benefit from reading a bit more into the literature that has provided a lot of insights or introduced more techniques onactivation sparsity, see e.g. [a-e]

[c] [MoEfication: Transformer Feed-forward Layers are Mixtures of Experts](https://arxiv.org/abs/2110.01786)

[d] [The Lazy Neuron Phenomenon: On Emergence of Activation Sparsity in Transformers](https://arxiv.org/abs/2210.06313)

[e] [Universal Properties of Activation Sparsity in Modern Large Language Models](https://arxiv.org/pdf/2509.00454)

[f] [Sparsing Law: Towards Large Language Models with Greater Activation Sparsity](https://arxiv.org/pdf/2411.02335)

[g] [LookupFFN: Making Transformers Compute-lite for CPU inference](https://arxiv.org/abs/2403.07221)

**Reviewer Concerns:**

Reviewer GaZW

- Novelty compared with other work: Addressed for TurboSparse and Q-Sparse, but misses comparison with other related ones such as [a, b].

- Loihi 2 sparse results are modeled, not measured: Acknowledged and cited other papers to support the validity of their performance model.

- Extra parameter size: Answered and addressed.

Reviewer TZy8

- The technical questions regarding the choice of $k=10$, the intuition behind Eq. 3, and the learning mechanism for $\Delta$: Addressed.

- Limited contribution and lack of evaluation against other state-of-the-art sparsification methods: The authors largely dismissed this as beyond the scope of the current work. This is unlikely to satisfy a reviewer who felt the contribution was already "very limited".

Reviewer R6HG

The reviewer explicitly replied to the rebuttal stating: "I feel I have given a high enough score and wont be changing it".

Reviewer JPIG

- Additional computational overhead and convergence time to be discussed: The authors provided specific metrics to address the concern.

- "actual measured results" or even data on a "single, small-scale sparse layer": The authors repeated the same response given to other reviewers, relying on literature references.

**Reviewer Scores:**

Reviewer GaZW

Remains 4. The request for measured hardware results which critical for a paper claiming specific energy/throughput gains on specific hardware remained theoretical.

Reviewer TZy8

Remains 2. Main concern was not addressed.

Reviewer R6HG

Remains 6. The reviewer explicitly replied to the rebuttal stating: "I feel I have given a high enough score and wont be changing it".

Reviewer JPIG

Remains 4. Main concern was not addressed.

---

### Decision · Program_Chairs · 2026-01-26

Reject